# Obsessive-Compulsive Disorder and Decision Making under Ambiguity: A Systematic Review with Meta-Analysis

**DOI:** 10.3390/brainsci11020143

**Published:** 2021-01-22

**Authors:** Veronica Nisticò, Andrea De Angelis, Roberto Erro, Benedetta Demartini, Lucia Ricciardi

**Affiliations:** 1Dipartimento di Scienze della Salute, Università degli Studi di Milano, 20142 Milano, Italy; veronica.nistico@unimi.it (V.N.); benedetta.demartini@unimi.it (B.D.); 2Aldo Ravelli Research Center for Neurotechnology and Experimental Brain Therapeutics, Università degli Studi di Milano, 20142 Milano, Italy; 3Neurosciences Research Centre, Molecular and Clinical Sciences Research Institute, St George’s University of London, London SW17 0RE, UK; andreadeangelis.md@gmail.com; 4Department of Neuropsychiatry, St George’s Hospital, South West London and St George’s Mental Health NHS Trust, London SW17 0RE, UK; 5Dipartimento di Medicina, Chirurgia e Odontoiatria “Scuola Medica Salernitana”, Università di Salerno, 84018 Baronissi, Salerno, Italy; rerro@unisa.it; 6Unità di Psichiatria II, Presidio San Paolo, ASST Santi Paolo e Carlo, 20142 Milano, Italy

**Keywords:** obsessive-compulsive disorder, decision making, Iowa Gambling Task, ambiguity, risk, meta-analysis

## Abstract

In the last decade, decision-making has been proposed to have a central role in obsessive-compulsive disorder (OCD) aetiology, since patients show pathological doubt and an apparent inability to make decisions. Here, we aimed to comprehensively review decision making under ambiguity, as measured by the Iowa Gambling Task (IGT), in OCD, using a meta-analytic approach. According to PRISMA Guidelines, we selected 26 studies for a systematic review and, amongst them, 16 studies were included in a meta-analysis, comprising a total of 846 OCD patients and 763 healthy controls (HC). Our results show that OCD patients perform significantly lower than HC at the IGT, pointing towards the direction of a decision making impairment. In particular, this deficit seems to emerge mainly in the last three blocks of the IGT. IGT scores in OCD patients under the age of 18 were still significantly lower than in HC. Finally, no difference emerged between medicated and unmedicated patients, since they both scored significantly lower at the IGT compared to HC. In conclusion, our results are in line with the hypothesis according to which decision making impairment might represent a potential endophenotype lying between the clinical manifestation of OCD and its neurobiological aetiology.

## 1. Introduction

Obsessive-compulsive disorder (OCD) is characterized by the presence of recurrent, persistent, intrusive thoughts (obsessions) and/or repetitive behaviours (compulsions) that an individual feels the urge and the need to perform, in order to prevent or reduce anxiety (DSM-5, [1]).

OCD affects approximately 2% of the population in the United States [2]. It is not a unitary disorder, but rather consists of different symptom clusters: a large meta-analysis [3], with a total sample size of 5124 patients, showed that a structure with four dimensions (symmetry, taboo thoughts, contamination and hoarding) could explain the heterogeneity of OCD symptoms. The most widely used measure of OCD symptoms is the Yale-Brown Obsessive-Compulsive Scale (Y-BOCS), a semi-structured interview whose results should reflect the average of the onset of each symptom throughout the week before the interview itself.

Several studies suggest that OCD is a familiar disorder and that genetic factors play a decisive role in its developing. Overall, it has been estimated that the heritability of OCD is around 40% and the remaining variation seems to be due to non-genetic factors, such as adverse perinatal events, psychological and neurological trauma, with a subsequent effect on the cortico-striato-talamo-cortico circuit [4].

Beside the known neurobiological and genetic mechanisms associated to OCD, it has been argued that OCD features might be better outlined by the study of endophenotypes [5,6]. Endophenotypes have been defined as intermediate phenotypes that are “not obvious or external but microscopic and internal” [7] thereby constituting heritable quantitative traits intermediate between disease phenotypes and the biological processes that underlie them [6,8].

Neuropsychological impairment is considered a potential endophenotype that lies between the clinical manifestation of OCD and its neurobiological aetiology [9]. However, cognitive domains in OCD have been investigated in numerous studies with inconsistent results [9].

Among neuropsychological domains, decision-making, defined as the process of selecting a particular option from a set of alternatives expected to produce different outcomes [10], has been proposed to have a central role in OCD [11], since patients show pathological doubt and an apparent inability to make decisions [9].

Furthermore, depending on the degree of uncertainty and the utility of information about future consequences provided to the subject, there are at least two main facets of decision making, namely decision making under ambiguity and decision making under risk. In decision making in ambiguous situation the information about contingencies, gains and losses is not clear, while in decision making in risky situation the participant is provided with explicit rules for rewards and punishments, and obvious probabilities [12].

Decision making under risk is mainly assessed by the Game of Dice Task (GDT) [13] and by the Cambridge Gambling Task (CGT). Regardless the task considered, there is inconsistency of results, with most studies not showing any correlation between decision making under risk and OCD [14,15,16,17,18,19,20].

The main way to assess decision making under ambiguity is the Iowa Gambling Task (IGT) [21]. In this card game, the subject’s objective is to win as much play-money as possible, or avoid losing as much as possible. The subject is required to make 100 card selections from four decks. Two decks (disadvantageous decks) are associated with high gains but with bigger losses so that they are disadvantageous overall. The other two decks C and D (advantageous decks) pay less but the penalties are lower, with an overall gain in the long run. The subjects are told that some decks are better than others and that they can switch between the four decks. After each choice, subjects are either given money or obliged to pay a penalty according to a programmed schedule of reward and punishment. This feedback is shown on the computer screen. The rules for gains and losses are implicit and unpredictable for every choice. Moreover, the subject does not know the exact number of remaining trials. Through the given feedbacks, subjects should learn to avoid disadvantageous decks to favour advantageous ones.

A number of studies have explored decision making under ambiguity, as measured by the IGT, in OCD, however there are discrepancies of results possibly related to methodological differences such as patients’ selection, symptoms heterogeneity or use of medications.

The aim of this study was to comprehensively review decision making under ambiguity, as measured by the IGT, in OCD, using a meta-analytic approach to synthesize the available data.

## 2. Materials and Methods

In September 2020, we conducted a systematic literature search in PubMed (http://www.ncbi.nlm.nih.gov) and EMBASE (https://www.elsevier.com/solutions/embase-biomedical-research) using the keywords “OCD” or “Obsessive-Compulsive Disorder” in combination with the keywords “Iowa Gambling Task” or “Gambling Task”. Strings inputted on PubMed, with respective results, are shown in the Appendix A. Additionally, the bibliographies of relevant articles were scanned for further suitable literature. Two authors (VN and ADA) screened all abstracts and full-text articles independently. Disagreement was resolved by discussion between the two independent authors; if no agreement was reached, a third independent party (either LR, BN or RE) was involved as an arbiter.

We included in the systematic review peer-reviewed articles containing data of original clinical studies on IGT performances in human subjects with OCD, compared to a group of healthy controls (HC), written in English. Duplicate articles, reviews, and opinions/comments were excluded. We screened the abstracts, full texts and Appendix A if available, for relevant information, which was then systematically entered into a comprehensive table. Twenty-six studies fulfilled the inclusion criteria and were therefore included in the qualitative synthesis. The review was conducted in accordance with the Preferred Reporting Items for Systematic Reviews and Meta-Analyses (PRISMA) statement [22].

Secondly, we conducted a meta-analysis of all the studies that provided sample size, means and standard deviations of the IGT scores. For studies in which a third group of patients with a psychiatric diagnosis other than OCD was investigated, we only included data on OCD patients and HC. In cases of missing data, authors were contacted and asked to provide the missing information. Data regarding patients who underwent surgery (such as deep brain stimulation) for their symptoms were excluded from the meta-analysis. Sixteen studies fulfilled the inclusion criteria; data of the included studies were pooled into a random-effects model, in line with the Cochrane Handbook for Systematic Reviews of Interventions (version 5.1.0) [23]. These results are indicated as standardized mean difference (SMD) with 95% confidence intervals (CIs). We additionally took under consideration of all the studies included in the pooled analysis to estimate clinical heterogeneity. This analysis was conducted using the I2 statistic, which provides an estimate of the percentage of inconsistency thought to be due to chance, along with the Chi2-*p* value that provides the strength of evidence for heterogeneity [24]. To explore reasons of heterogeneity, sensitivity analyses were performed considering two main biological factors potentially able to influence the outcome, i.e., in medicated and unmedicated patients, as well as in adults and in children (respectively above and below 18 years of age). The problem of publication bias (i.e., the existence of unpublished studies with negative results) was estimated informally by inspecting the funnel plot of effect size against standard error for asymmetry. We first run the analyses including the entire set of studies and then subsequently re-run them without the potential outliers identified based on visual inspection of the funnel plot.

Finally, in order to compare the learning curves of OCD and IGT throughout the five IGT blocks, we additionally performed meta-analyses of the subscores in each block. These analyses were performed with RevMan, version 5.3 [25].

## 3. Results

### 3.1. Systematic Review

A total number of 63 papers were screened and 37 were excluded for the following reasons: two were not in English; three were not original research articles (being either reviews or meta-analysis); four were not on human subjects; 13 were not about obsessive-compulsive disorder; nine were not about IGT; six did not include a healthy control group of comparison. Therefore, 26 articles were included in the data analysis, all published between 2002 and 2020 (Figure 1). Out of 26 studies, 13 (48.15%) compared IGT performances between OCD and HC, whereas 14 studies (51.85%) compared OCD cases with both HC and with patients with other psychiatric diagnosis, namely panic disorder [26], eating disorders [27], substance and alcohol abuse/dependence [28], pathological gambling [28,29], compulsive hoarding [30], attention deficit hyperactivity disorder (ADHD) [31] and schizophrenia [32].

#### 3.1.1. Performance at IGT

Overall, 18 studies found that OCD patients performed worse on IGT compared to control groups [15,16,18,19,20,26,29,30,32,33,34,35,36,37,38,39,40,41], whereas eight did not confirm this result [27,28,31,42,43,44,45,46]. Detailed findings of the included studies are reported in Table 1.

Moreover, Grassi and colleagues [36], analysed different sub-scores of the IGT, namely the net-score of blocks 1 + 2 and the net-score of blocks 3 + 4 + 5 as measures of decision-making under ambiguity and under risk respectively. They showed that the overall net score was significantly worse in OCD patients than in HC but there was no significant difference between OCD patients and HC in decision-making either under ambiguity or risk although healthy controls’ performance significantly improved from the first block to the last block of choices while patients’ performance did not. Other studies [29,35,37] found a significant difference between OCD patients and HC in decision making under risk but not under ambiguity.

#### 3.1.2. Correlation with Symptoms and Severity

Overall, IGT scores were not related with symptomatology and illness severity, as measured by the Y–BOCS scale, in the majority of studies [20,26,35,43,44]. However three studies found a correlation between symptoms severity as per Y-BOCS and IGT scores [18,39,45]. Kim and colleagues reported a significant correlation between IGT scores and depressive symptoms measured with the Montgomery-Asberg Depression Rating Scale, while Da Rocha and colleagues [35] did not find any correlation with depressive or anxiety symptoms (as per Beck Anxiety Inventory and Beck Depression Inventory).

#### 3.1.3. OCD Clusters

Although Laurence and colleagues [44] did not find a difference in IGT scores between HC and OCD overall, they showed that OCD patients with a high score on a hoarding subscale (10 subjects, identified with the Y-BOCS, the Obsessive-Compulsive Inventory-Revised (OCI-R) Hoarding Subscales, and the Savings Inventory-Revised) scored significantly lower than other low-hoarding-OCD patients and than HC. However, direct comparison between compulsive hoarding patients and OCD patients in their performance at the IGT brought to controversial results in other studies. In Blom et al. study [30], compulsive hoarding patients’ showed a similar learning curved to healthy controls, while OCD group exhibited slower learning and a lower total net score overall. On the other hand, Tolin and Villavicencio [46] found no group difference between patients with hoarding disorder, patients with OCD and HC nor at the overall net score, nor at the learning trends across blocks. Finally, Martoni and colleagues [40], not only found that OCD patients scored significantly lower than HC at the IGT, but shed light on a different behaviour within OCD-subtypes. Specifically, IGT performance of high-hoarding-OCD patients (as well as patients with “rituals” and “forbidden thoughts” as prevalent symptomatology) did not improve over time; on the contrary, OCD patients with high scores in “washing” and “symmetry” subscales improved over time.

In Starcke and colleagues [16], only those three patients who reported symptoms of trichotillo-mania in addition to OCD performed worse on the IGT.

#### 3.1.4. Differences between OCD and Other Behavioural Addictions in IGT Performance

Few studies have evaluated the difference between OCD and other behavioural addictions, such as pathological gambling and alcohol dependents in IGT performance and they did not find any difference [28,29].

#### 3.1.5. Gender

Most of the studies which investigated gender effect found neither a significant association between the IGT net score and gender, nor a difference in IGT performance between male and female subjects [15,18,32,35]. Only Lawrence and colleagues [44] found a significant difference at the IGT performance between male and female participants, with 69% of men and 34% of women guessing the correct (or partially correct) strategy, but they found neither a difference between HC and OCD patients, nor an interaction effect between gender and diagnosis. Finally, Martoni and colleagues [40] found that, within patients scoring higher at the “forbidden thoughts” factor of the Y-BOCS, female patients at had a lower probability to provide correct answers in the IGT test than male patients.

#### 3.1.6. Pathological Doubt

Only one study [27] evaluated the presence of a correlation between pathological doubt, as measured by the subscale “doubts about actions” of the Frost Multidimensional Perfectionism Scale, and the IGT score. They found that, only in OCD patients, IGT performance on the last two blocks was positively associated with the degree of doubts about actions.

### 3.2. Meta-Analysis

Out of the 26 articles included in the systematic review, 16 articles were eligible to be included in the meta-analysis, comprising a total of 846 OCD patients and 763 HC.

One study [19] provided separate IGT data about medicated and unmedicated patients, and we included in our main meta-analysis only medicated subjects due to their higher numerosity. The group of unmedicated patients has been instead included in the sub-analysis, pooling data of unmedicated patients, as described below.

The main analysis showed that the IOWA score was significantly lower in OCD patients than in HC (SMD = −0.65 [95% CI: −0.76; −0.55], *p* < 0.001; Figure 2). However, there was considerable heterogeneity among the included studies [I2 = 84% *p* < 0.001].

The funnel plot (Figure 3) was fairly symmetrical suggesting a low risk of publication bias. However, five studies overall accounting for 22.7% of the total weight in the meta-analysis were identified as possible outliers (Figure 2 and Figure 3).

A sensitivity analysis was performed after removal of these 5 studies, which confirmed a significant difference in the IGT between OCD and HC (SMD: −0.64 95% C.I. [−0.76; −0.52]; z = 10.54; *p* < 0.001) (Figure 4). In this case, however, the heterogeneity between studies dropped significantly (I-2 = 2%) and was no longer significant (chi-2 = 10.18; *p* = 0.42).

#### 3.2.1. Sensitivity Analysis

##### Age

Overall, OCD and HC groups were balanced for age, thus it is not possible to draw general conclusions about the influence of age on the different performances achieved at the IGT. Two studies [31,39] included patients below 18 years old. However, meta-analysis results did not change by excluding these two papers: specifically, IOWA score in adult patients with OCD was still significantly lower than in adult HC (SMD = −0.68 [95% CI: −0.98; −0.39], *p* < 0.001; Figure 5). Heterogeneity remained considerable [I2 = 86%; *p* < 0.001].

When considering only the two studies including children, a significant difference between OCD patients and HC performances at the IGT emerges, with OCD performing lower than HC (SMD = −7.17 [95% CI: −12.05; −2.28], *p* = 0.004; Figure 5). As expected, CIs were larger, but no significant heterogeneity was observed [I2 = 0%; *p* = 0.81].

##### Medications

Amongst the selected studies, psychotropic medications were given in 13 studies to at least a part of the patients’ group, while the other three studies enrolled only drug-naïve patients [43] or patients who were drug-free since at least two weeks [26,32].

Two sub-analyses have been run including studies focusing on medicated and drug-free/naïve patients, respectively.

IGT score in medicated patients was significantly lower than in HC (SMD = −0.56 [95% CI: −0.72; −0.40], *p* < 0.001; Figure 6). Heterogeneity was significant, but considerably lower than in the general meta-analysis [I2 = 45%; *p* = 0.04].

IGT score in unmedicated patients was still significantly lower than in HC (SMD = −1.06 [95% CI: −2.06; −0.06], *p* = 0.04; Figure 6). In this case, heterogeneity remained considerable [I2 = 95%; *p* < 0.001].

#### 3.2.2. IGT Blocks

Five studies [16,19,28,36,44] out of 16 reported data of the net score obtained by HC and OCD in each of the 5 IGT blocks. Data are summarized in Figure 7.

Meta-analysis showed that scores were not significant different in OCD compared to HC in block 1 (SMD = −0.09 [95% CI: −0.28; 0.09], *p* = 0.33; I2 = 0, *p* = 0.45) and block 2 (SMD = −0.20 [95% CI: −0.50; 0.11], *p* = 0.21; I2 = 59%, *p* = 0.21), whereas significant lower scores were observed for OCD patients in block 3, (SMD = −1.06 [95% CI: −1.92; −0.20], *p* = 0.02), block 4 (SMD = −0.97 [95% CI: −1.75; −0.19], *p* = 0.01) and block 5 (SMD = −0.79 [95% CI: −1.34; −0.24], *p* < 0.01). However, heterogeneity amongst the studies in the analysis for block 3, 4 and 5 was relevant [I2 = 94%, *p* < 0.01; I2 = 91%, *p* < 0.01; I2 = 86%, *p* < 0.01 respectively].

## 4. Discussion

This systematic review and meta-analysis aimed to summarize the available data in the literature on the ability of decision making in patients with OCD assessed with the Iowa Gambling Task.

The majority of studies (66.67%) showed a significant decision-making impairment in patients with OCD. This finding suggests that the impairment of decision-making in OCD might constitute a trait feature of the disorder itself and a potential endophenotype [9]. It has been hypothesized that individuals with OCD present a significant impairment in decision making in the context of obsessive doubting and uncertainty [19,47]. In fact, pathological doubt (i.e., a lack of certitude or confidence in one’s memory, attention, intuition, and perceptions, such that it is difficult to trust one’s internal experiences [48,49]) is central in the clinical presentation of OCD. It underpins several symptoms, from “checking behaviors” (insufficient conviction about the completion of a task) to contamination concerns (insufficient conviction regarding the safety of a contacted object) [48], and this pervasive lack of certainty leads OCD patients towards a significant impairment in their daily life functioning. In addition to clinical observation, several experimental studies have investigated the chronic doubting and indecision characteristic of OCD [49] (for a discussion see [48]), which have been ultimately attributed to decision-making impairments [38,50]. Thus, previous literature considered decision making at the basis of obsessive and compulsive symptoms [47] and a possible candidate for a potential endophenotype. With an estimated mean score 0.65 points lower than healthy controls, our results support the notion of impaired decision-making under ambiguity in OCD patients compared to HC as measured with the IGT. Moreover, it must be noticed that the differences in the net scores between the two groups is significant especially in the last three blocks, with OCD patients scoring lower than HC. Given the structure of the IGT, some authors suggested that the choices in the last blocks resemble more a task of decision-making under risk than under ambiguity [18,36]. In fact, at the time of approaching block 3, participants should have understood the rules of the game and therefore should be able to pick up cards from the more advantageous decks. Following this hypothesis, our results would point towards the direction of a deficit in decision-making under risk, more than under ambiguity in OCD, in line with several studies included in our systematic review [18,19,29,35,37]. However, few studies [16,18,19] have compared the performance of OCD patients and HC both at the IGT and at the Game of Dice Task (a task directly assessing decision making under risk) and they showed that, although OCD patients performed lower than HC at the IGT, no difference emerged between the two groups at the GDT, suggesting no impairment in making decision under risk.

Another explanation arose when considering the trend of choices of the two groups, with controls’ performance significantly improving from the first block to the last block of choices while patients’ performance did not—again suggesting OCD patients’ difficulties in learning or developing an adequate strategy [36]. It has been suggested that OCD patients might have a deficit in reversal learning, the ability to inhibit or suppress a response when it is no longer rewarding [51,52,53], however, in another study, OCD patients did not show a lower performance than HC at a Simple Reversal Learning Task, and the number of reversal errors did not correlate with IGT scores [18] thus suggesting that the reduced IGT performance might not be due to reversal learning deficit.

To explain heterogeneity among studies, we attempted to evaluate the influence of age on IGT performance by pooling studies considering subjects under the age of 18 [31,39] and studies evaluating subjects above the age of 18. Although it was not possible to draw general conclusions about the influence of age, given that results were significant in both age groups, it emerged that differences in IGT scores in adolescents with and without OCD were larger as compared to those observed in adults. To the best of our knowledge, only two studies evaluated IGT performance in adolescents with OCD: Kodaira and colleagues [39] found that children with OCD performed similar to controls in the early stages of the IGT, but selected more disadvantageous cards at the very last block; this behaviour has been observed in patients with a deficit of the orbitofrontal cortex. In fact, Norman and colleagues [31] by performing functional magnetic resonance imaging during the IGT, found that adolescents with OCD showed disorder-specific under-activation in ventral-medial-orbitofrontal cortex during advantageous choices; however, since they used a shortened version of the IGT, it was not possible to assess whether their sample would have showed the same trend as Kodaira et al. [39] at the IGT net score. On the other hand, several studies have compared IGT performance between healthy adolescents and adults and found that younger participants made more disadvantageous choices relative to older participants [54,55]. In particular, adolescents and children were found to prefer the advantageous decks during the entire task, while adults progressively developed the strategy of refraining from playing from the disadvantageous decks [52]. The authors suggested, in line with previous studies [55,56,57] that the late maturation of the orbitofrontal “control system” might explain their finding of a peak in reward sensitivity during adolescence at the IGT, as well as in other laboratory-tested decision-making tests involving emotional and social, rather than cognitive, factors [57,58,59]. We might speculate, therefore, that the disorder-specific under-activation in the ventral-medial-orbitofrontal cortex found by Norman and colleagues in OCD patients [31] might explain their worse IGT performance and ultimately account for the wider SMD occurring between adolescents with and without OCD, compared to the SMD between adults with and without OCD. Finally, inhibitory control was shown to be impaired in unmedicated adolescent patients with OCD, and this impairment was correlated with the severity of their symptoms [60]. Inhibitory control is part of the executive function domain, which allows adapting motor behavior according to the context in which the subject is embedded; in particular, reactive inhibition refers to the ability of the person to react to a stop signal, and it is usually evaluated through a Stop Signal Reaction Time Task (SSRTT); proactive inhibition consist in the ability of patients to shape their response strategies in anticipation of known task demands [60,61]. Mancini and colleagues [60] found that the more severe were the OCD symptoms, the more impaired were both proactive and reactive inhibition in a group of adolescents with OCD (but not in an age-matched group of patient with Tourette syndrome). Decision making, and thus the IGT, is strongly affected by one’s ability of response inhibition [62]. In our systematic review, only one paper implemented both the IGT and the SSRTT in a sample of adult with OCD and HC, and found that OCD performed worse than HC at the IGT but not at the SSRTT [30]; future studies investigating proactive inhibition in adults with OCD might help clarify whether this phenomenon occurs only in children and adolescent with OCD or could be generalized to the adult age.

Moreover, we further took into account the effect of medications, by performing sub-analyses in medicated and unmedicated patients; it emerged that both medicated and unmedicated patients scored significantly lower at the IGT, compared to HC, with the size effects being similar. Several studies included in our systematic review [16,27,28,36] addressed the same question, and none of them found a significant effect of any class of drugs on IGT performances. On the other hand, Cavedini et al. [26] reported that patients who did not respond to selective serotonin-reuptake inhibitors (SSRIs) showed impairments on the IGT, while patients responding to pro-serotonergic treatment played as well as healthy controls did. Finally, Zhang, et al., [19] compared IGT performance in a group of patients with active OCD, people with past OCD in remission at the time of the study and HC, and reported that, regardless the medication status of patients, deficits in decision making under ambiguity existed and remained unchanged even despite symptoms’ remittance.

Finally, as mentioned above, OCD is a heterogeneous disorder, which might be better understood by considering its sub-clusters. In particular, in our systematic review, it emerged that high-hoarding-OCD patients perform worse than both HC and low-hoarding OCD patients in decision-making tasks [40,44]. However, when directly comparing the IGT performance of patients with compulsive hoarding and patients with OCD, controversial results emerged: one study [30] found that OCD patients were still scoring significantly lower than compulsive hoarding patients, and another study [46] found no difference between patients with hoarding disorder, patients with OCD and HC. Therefore, although it is not possible to draw firm conclusions, we might hypothesize that patients with a primary diagnosis of hoarding disorder have different cognitive endophenotypes compared to OCD patients with additional hoarding symptoms.

Overall, our results should be interpreted cautiously as a significant heterogeneity among studies emerged in most of the performed meta-analyses. Five studies [26,32,42,43,44] contributed the most to heterogeneity. The factor that seemed to have the greater influence on the heterogeneity of the main meta-analysis was the medical treatment. Indeed, considering only medicated patients, I2 index of heterogeneity dropped from 86% of the main analysis to 45%. This might be possibly due the inclusion in the main analysis of medication-naïve [43], medication-free [26,32] and medicated patients [16,18,19,28,29,31,35,36,37,38,39,42,44], which might suggest that in fact a possible role of medications on this task. It is worthwhile to note, in fact, that the study by Cavedini et al. [26] and the one by Cavallaro et al. [32] strongly contributed to the heterogeneity in our main meta-analysis, showing the greatest individual differences. Additionally, these two studies showed confidence intervals wide enough to suggest high variability within the single study. Finally, given that OCD is a multifaceted disorder which encompasses several domains, and that not all the authors pointed out the subtype of OCD tested in their researches, the heterogeneity observed amongst the studies included in our meta-analysis might just reflect the heterogeneity of OCD itself.

The mean IGT difference between OCD and HC was −0.65 in our meta-analysis, and this might seem a low value to be clinically significant; however, our standardized mean difference (SMD) is in line with the SMD of other meta-analysis reporting a relevant clinical difference at the IGT between healthy controls and other neuropsychiatric populations [63,64,65,66]. Most importantly, the SMD resulting from our meta-analysis is close to the one of a meta-analysis comparing HC and pathological gamblers (d = −1.03) [63]. Since pathological gambling is the prototypical disorder where decision making under ambiguity and under risk are impaired and IGT is particularly sensible in detecting this phenomenon, the similar results obtained in OCD and pathological gambling support the relevance of the phenomenon we observed.

We acknowledge the limitations of our study. First, several studies included in the systematic review could not be included in our meta-analysis, as they did not report the necessary statistical data; for the same reason, we could not run a meta-analysis evaluating a possible effect of gender and age of onset, since most of the study did not investigate gender and age of onset effect and thus did not report the relevant statistical data; second, few studies reported precise OCD symptomatology (cluster), which might account for the heterogeneity in our meta-analysis; third, few studies assessed decision making impairment in OCD under the age of 18 and differences between medicated and unmedicated patients; finally, amongst medicated patients, it was not possible to further divide the sample according to the type of medication taken due to the small sample sizes, which might have create another bias.

In conclusion, our results show an impairment in decision making, in particular under risk, in OCD patients compared to HC. This finding suggests that the neuropsychological domain of decision making might represents a possible endophenotype accounting for OCD symptomatology.

## Figures and Tables

**Figure 1 brainsci-11-00143-f001:**
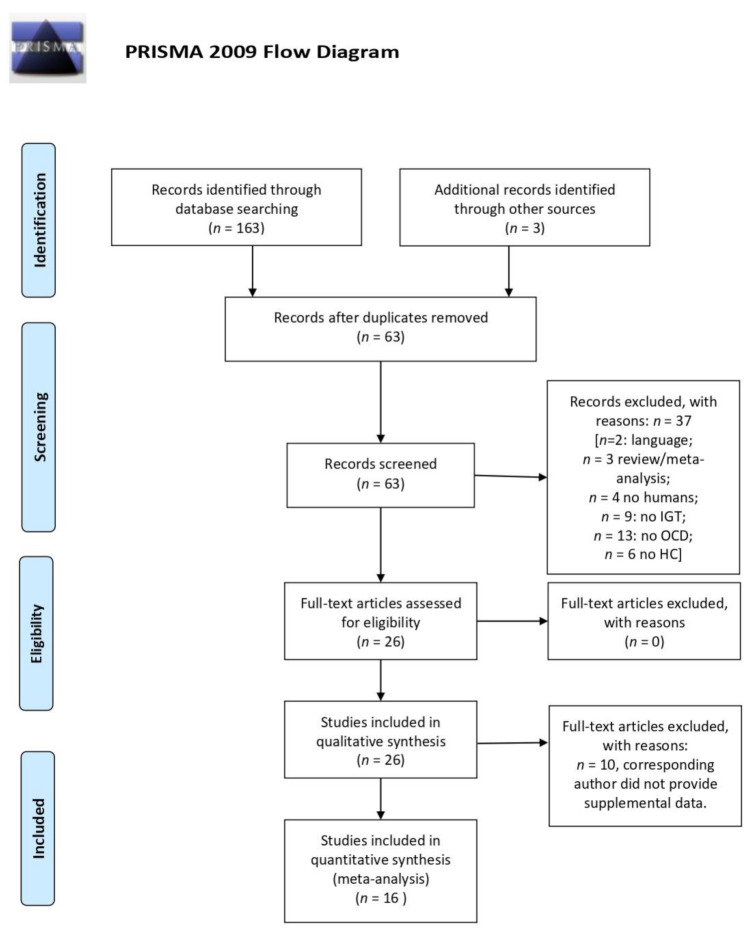
PRISMA flow diagram: Systematic Review and Meta-analysis method.

**Figure 2 brainsci-11-00143-f002:**
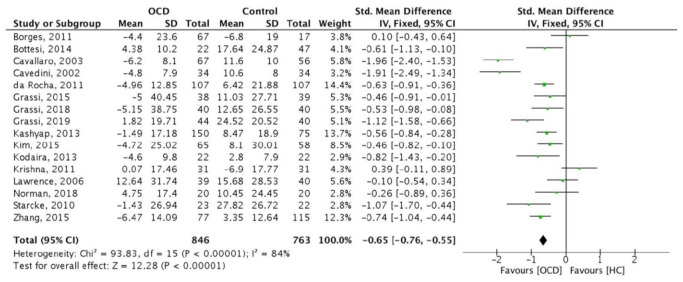
Forest plot of the 16 studies included in the meta-analysis.

**Figure 3 brainsci-11-00143-f003:**
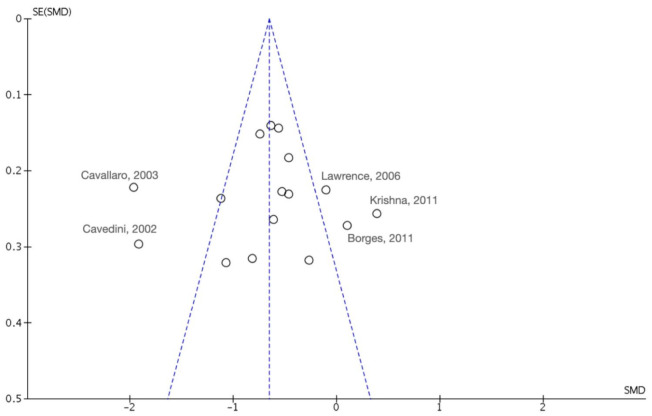
Funnel plot of effect size against standard error with identification of potential outliers.

**Figure 4 brainsci-11-00143-f004:**
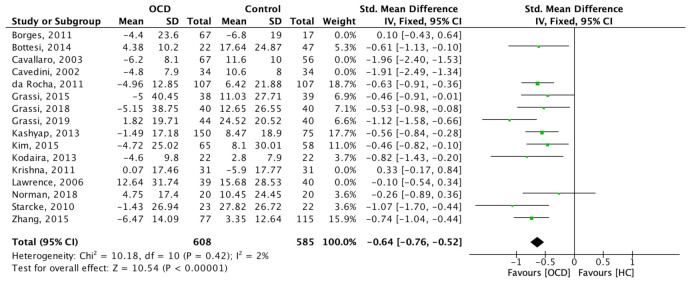
Forest plot of the main meta-analysis after removing potential outliers.

**Figure 5 brainsci-11-00143-f005:**
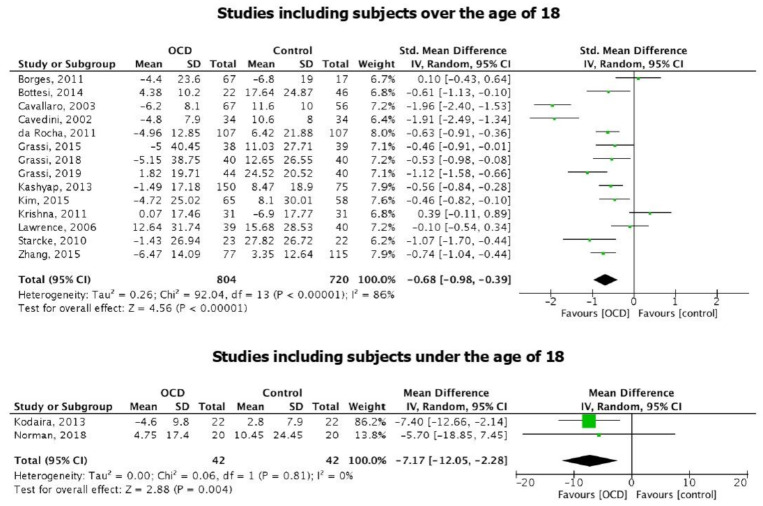
Forest plots of the studies including subjects over the age of 18 and the studies including subjects under the age of 18.

**Figure 6 brainsci-11-00143-f006:**
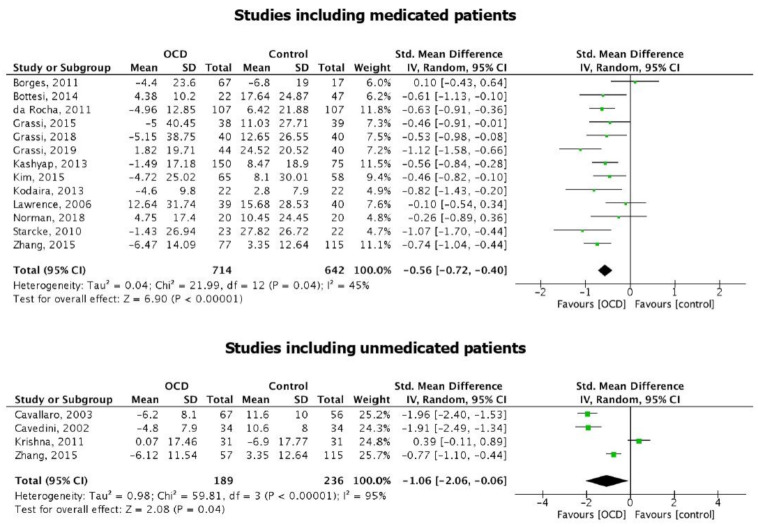
Forest plot of the studies including medicated patients and the studies including unmedicated patients.

**Figure 7 brainsci-11-00143-f007:**
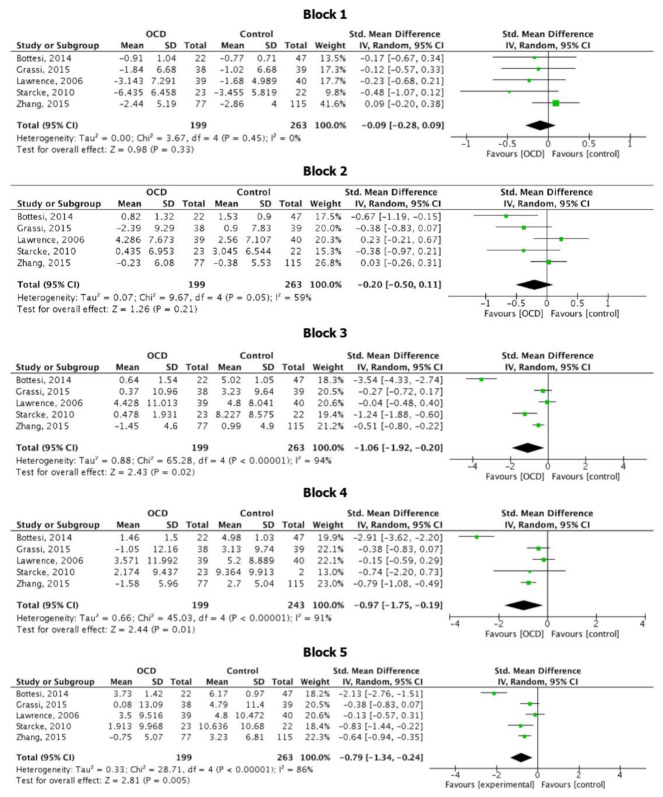
Forest plot relative to the IGT score at each block.

**Table 1 brainsci-11-00143-t001:** Studies included in the systematic review.

First Author, Year [Reference]	Included in Meta-Analysis?	Experimental Groups	Age	Gender	Disease Duration	N° of Patients Receiving Medication	Method—Diagnosis and Psychiatric Assessment	Method—Neuropsychological Assessment	IGT Findings	Medication Effect
**Blom et al., 2011** [30]	no	OCD non Hoarding (*n* = 17) vs. HC (*n* = 19) vs. HD (*n* = 24, of whom 14 with past OCD)	43 [range: 19–63]	Majority F	N/A	HD: 13; OCD: 16	Clinical interview (DSM-IV criteria); OCI-R	SI-R, The vocabulary scale from the Shipley-2, SRTT, SSRTT, IGT	OCD < HC OCD < HD	N/A
**Boisseau et al., 2013** [27]	no	OCD (*n* = 19) vs. ED (*n* = 17) vs. HC (*n* = 21)	OCD: 22.32 ± 4.24, ED: 23.12 ± 4.80, HC: 24.24 ± 3.47	F = 100%	OCD: 6.74 ± 4.92, Eds: 6.79 ± 4.95	OCD: 5, ED: 3	Clinical interview (DSM-IV criteria); Y-BOCS and Y-BOCS-SC, EDE-Q, FMPS, MINI	IGT	No significant differences between OCD and HC.	No medication effect
**Borges et al., 2011** [42]	yes	pst-OCD (*n* = 16) vs. prt-OCD (*n* = 18) vs. ntOCD (*n* = 67) vs. HC (*n* = 27)	Pst-OCD: 39.2 ± 12.4, Prt-OCD: 41.2 ± 12.3, NonT-OCD: 33.0 ± 13.2, HC: 29.9 ± 7.9	Pst-OCD: F = 9, M = 7; Prt-OCD: F = 11, M = 7; NonT-OCD: F = 39, M = 28; HC: F = 12, M = 5	N/A	Most patients were under psychological and/or pharmacological treatment	Clinical interview (DSM-IV criteria), SCID-I for DSM-IV criteria, Y-BOCS, BDI, BAI	WCST, IGT, WMS-R LM, BVMT-R, WASI	No significant differences between OCD and HC.	N/A
**Bottesi et al., 2014** [28]	yes	PG (*n* = 40) vs. AD (*n* = 40) vs. OCD (*n* = 22) vs. HC (*n* = 47)	PG: 40.01 ± 12.05, AD: 31.27 ± 11.58, OCD: 47.15 ± 10.43, HC: 43.06 ± 11.94	PG: F = 5%, M = 95%, AD: F = 27.3%, M = 72.7%, OCD: F = 50%, M = 50%, HC: F = 21.3%, M = 78.7%	N/A	PG: 17, AD: 33, OCD: 12	Y-BOCS, BDI-II, BAI, SOGS, PSWQ	AUDIT, PI, OBQ-87, BIS-11, the Go/No-go task, IGT	No significant differences between OCD and HC, nor between PG and OCD.	No medication effect
**Cavallaro et al., 2003** [32]	yes	OCD (*n* = 67) vs. SKZ (*n* = 110) vs. HC (*n* = 56)	OCD: 30.5 ± 8.9, SKZ: 33 ± 9.5, HC: 31.2 ± 6.0	OCD: F = 50.8%, M = 49.2%, SKZ: F = 40%, M = 60%, HC: F = 60.8%, M = 39.2%	OCD: 10.4 ± 8.1; SKZ: 9.4 ± 7.2	all patients: medication-free for at least 2 weeks.	Clinical interview (DSM-IV criteria)	the Gambling Task, WCST, ToH	OCD < HC OCD < SKZ	N/A
**Cavedini et al., 2002** [26]	yes	OCD (*n* = 34) vs. PD (*n* = 16) vs. HC (*n* = 34)	OCD: 33.7 ± 11.5, PD: 36.3 ± 10.9, HC: 29.5 ± 8.9	OCD: F = 47%, M = 53%, F = 56.2%, M = 43.8%, HC: F = 55.8%, M = 43.2%	N/A	all patients: medication-free for at least 2 weeks.	Diagnostic Interview Schedule III R, RY–BOCS	IGT	OCD < HC. OCD patients made significantly more selections from the disadvantageous decks than PD and HC.	Poor neuropsychological task performance predicted poor outcome of pharmacological treatment
**Cavedini et al., 2010** [33]	no	OCD probands and UFDR (35 pairs) vs. HC (31 pairs)	OCD probands 35.6 ± 2.9, relatives 45 ± 3, HC:probands 34.7 ± 2.9, relatives 43.2 ± 2.5	OCD: probands F = 15, M = 20, relatives F = 22, M = 13, HC: probands F = 22, M = 9, relatives F = 23, M = 8	N/A		Clinical interview (DSM-IV criteria); Y-BOCS, MINI-DIS	Y-BOCS, IGT, ToH, WCST	OCD < HC. HC probands chose more frequently from the advantageous decks than OCD probands.	N/A
**Cavedini et al., 2012** [34]	no	OCD (*n* = 20) vs. HC (*n* = 18)	OCD: 36.05 ± 11.05, HC: 27 ± 4.73	OCD: F = 7, M = 13, HC: F = 5, M = 13	N/A	all patients: medication-free for at least 1 month	Clinical interview (DSM-IV-TR criteria), Y-BOCS, MINI-PLUS	IGT, SCR	OCD < HC. OCD showed no significant differences of SCRs activation according to card selections, while HC did.	N/A
**da Rocha 2011** [35]	yes	OCD (*n* = 107) vs. HC (*n* = 107)	OCD: 28.40 ± 14.12, HC: 29.33 ± 13.22	OCD: F = 49, M = 58, HC: F = 51, M = 56	Average duration of illness 111.54 ± 94.36 months; Average duration of untreated illness 78.40 ± 43.01 months	OCD: 85	MINI-PLUS interview, Y-BOCS, DY-BOCS, BID, BAI, review of medical records, interview with minimum 1 close relative	Raven Progressive Matrices, IGT, CPT-II	OCD < HC	N/A
**Grassi et al., 2015** [36]	yes	OCD (*n* = 38) vs. HC (*n* = 39)	OCD: 36.29 ± 12.73, HC: 34.10 ± 11.18	OCD: F = 39.47%, M = 60,53%, HC: F = 51.28%, M = 48.72%	N/A	OCD: 32	SCID-I for DSM-IV criteria; Y-BOCS, Y-BOCS-SC	BIS-11, IGT, the beads task	OCD < HC. OCD did not improve across the blocks. No difference between groups in performance under ambiguity and under risk emerged.	No medication effect
**Grassi G, 2018** [37]	yes	DBS-OCD (*n* = 20) vs. TAU-OCD (*n* = 40) vs. HC (*n* = 40)	DBS-OCD: 45.65 ± 12.7, TAU-OCD: 44.75 ± 11.5, HC: 44.08 ± 9.96	DBS-OCD: F = 55%, M = 45%; TAU-OCD: F = 45%, M = 55%; HC: F = 52.5% M = 47.5%	DBS-OCD: 26.5 [range: 20–38]; TAU-OCD: 30 [range: 17–35]	20 OCD treated with DBS targeting the ventral limb of the internal capsule, 40 OCD TAU: medication and/or CBT	SCID-I and SCID-II for DSM-IV criteria; Y-BOCS, Y-BOCS-SC	IGT, the beads task.	OCD < HC	No differences were found between OCD patients treated with DBS or TAU
**Grassi G, 2019** [29]	yes	OCD (*n* = 44) vs. GD (*n* = 26) vs. HC (*n* = 40)	OCD: 33 [range: 42.75; 26]; GD: 39 [57.25; 33.75]; HC: 34 [48; 27]	OCD: F = 11.5%; GD: F = 31.8%; HC: F = 20%	OCD: 15.6 ± 10.4; GD: 12.1 ± 9.9	OCD: 93.2%; GD: 61.5%	SCID-I and SCID-II for DSM-IV criteria; Y-BOCS, Y-BOCS-SC, PG-YBOCS, HDRS, HARS, SHAPS	TIB, BIS-11, IGT, FTND, Burghart’s Sniffin’ Sticks Screening Test	OCD < HC GD < HC OCD = GD	N/A
**Kashyap et al., 2013** [38]	yes	OCD (*n* = 150) vs. HC (*n* = 205 of whom 75 performed the IGT)	OCD: 27.56 ± 7.35, HC: 27.42 ± 6.57.	OCD: F = 56, M = 94; HC: N/A	8.39 ± 5.69	80% of OCD: SRIs. 29.9% CBT. 44.5% augmentation, either with an antipsychotic (13.9%) or a benzodiazepine (25.5%)	Clinical interview (DSM-IV criteria), MINI Plus, Y-BOCS, CGI, BABS, STAI, HDRS- 17.	CTT, Digit Span (WMS III), Matrix Test (WAIS III), AVLT, CFT, ToH, WCST, OAT, IGT, SCWT, COWA, Five-point Test, Verbal N-Back, Spatial Span, BGT	OCD < HC	N/A
**Kim et al., 2015** [18]	yes	OCD (*n* = 65) vs. HC (*n* = 58)	OCD: 26.62 ± 9.12, HC: 26.56 ± 6.28	OCD: F = 14, M = 51, HC: F = 22, M = 36	9.57 ± 7.46	OCD: 63	SCID for DSM-IV criteria. Y-BOCS, MADRS	IGT, GDT, SRLT, WCST	OCD < HC at the netscore and at the last three blocks.	N/A
**Kodaira et al., 2013** [39]	yes	OCD (*n* = 22) vs. HC (*n* = 22)	OCD: 163.5 ± 22.1 months, HC: 161.8 ± 20.6 months	OCD: F = 10, M = 12, HC: F = 10, M = 12	23.9 ± 21.3 months	OCD: 10	Clinical interview (DSM-IV-TR criteria), NIMH-OCS, CY-BOCS, IGT, WISC-III, WCST	IGT, WISC-III, WCST	OCD < HC. OCD selected a higher number of disadvantageous cards than HC in the last block; this number was associated with CY-BOCS score.	N/A
**Krishna et al., 2011** [43]	yes	OCD (*n* = 31) vs. HC (*n* = 31)	OCD: 26.0 ± 6.1, HC: 26.3 ± 6.2.	OCD: F = 7, M = 24, HC: F = 7, M = 24	48 months	all patients: medication-naïve	Clinical interview (DSM-IV criteria), MMSE, MINI, YBOCS	WCST, CPT, TMT, TOL, COWA, WMS, IGT, RCFT, OAT, BGT, fMRI, DST, Design fluency test, SCWT, Rey’s AVLT, Matrix reasoning test	No significant differences between OCD and HC.	N/A
**Lawrence et al., 2006** [44]	yes	OCD (*n* = 39) vs. HC (*n* = 40)	OCD: 36.1 ± 11.07, HC: 33.48 ± 10.4	OCD: F = 19, M = 20, HC: F = 20, M = 20	20.7 ± 12	OCD: 30	SCID I and II for DSM-IV criteria; Y-BOCS, Y-BOCS-SC, SI–R, BDI, STAI	IGT, WCST, NART, SCR	No significant differences between OCD and HC. IGT performance and SCR were impaired in patients with prominent hoarding symptoms.	N/A
**Martoni et al., 2015** [40]	no	OCD (*n* = 269) vs. HC (*n* = 120)	OCD: 34.43 ± 11.70, HC: 32.14 ± 11.02	OCD: F = 141, M = 129, HC: F = 78, M = 42	12.68 ± 9.98	OCD: 58.4%	Clinical interview (DSM-IV-TR criteria), Y-BOCS	WAIS-R, IGT	OCD < HC. Patients with high scores in ‘Washing’ and ‘Symmetry’ factors showed an improving performance across the blocks; patients with high scores in ‘Hoarding’, ‘Rituals’ and ‘Forbidden thoughts’ did not.	N/A
**Nielen et al., 2002** [45]	no	OCD (*n* = 27) vs. HC (*n* = 26)	OCD: 34.9 ± 9.9, HC: 31.2 ± 8.3	OCD: F = 20, M = 7, H: F = 18, M = 8	18.4 ± 12.3	all patients: medication-free	Clinical interview (DSM-IV-TR criteria), Y-BOCS, HARS, HDRS	the Gambling Task, RSPM	No significant differences between OCD and HC Within OCD group, IGT performance was associated with anxiety and OCD severity.	N/A
**Norman LJ, 2018** [31]	yes	OCD (*n* = 20) vs. ADHD (*n* = 16) vs. HC (*n* = 20)	HC: 15.15 ± 1.99, ADHD: 14.61 ± 1.87, OCD: 15.76 ± 1.43	M = 100%	N/A	OCD: 16 medication naïve, 4 SSRI, 1 risperidone augmentation treatment	clinical interview with patients and parents (following ICD-10 criteria); CY-BOCS	WASI-R, IGT, fMRI, SDQ	No significant differences between OCD and HC.	N/A
**Starcke et al., 2009** [15]	no	OCD (*n* = 14) vs. HC (*n* = 15)	OCD: 36.36 ± 8.54, HC: 36.60 ± 10.84	OCD: F = 7, M = 7, HC: F = 5, M = 10	N/A	OCD: 9	SCID for DSM-IV criteria	LPS-4, IGT, GDT, SCR	OCD < HC. HC showed higher SCR elevations after losses than after gain, OCD did not	N/A
**Starcke et al., 2010** [6]	yes	OCD (*n* = 23) vs. HC (*n* = 22)	OCD: 35.25 ± 7.35, HC: 36.50 ± 10.23	OCD: F = 10, M = 13, HC: F = 10, M = 12	N/A	OCD: 14	SCID for DSM-IV criteria	GDT, IGT, AFLT, LPS-4, mWCST, TMT A and B, ToH, F-A-S-test, the Word Color Interference Test.	OCD < HC at netscore, block 3, block 5.	No significant differences for medicated and unmedicated patients on any of the decision-making tasks
**Tolin and Villavicencioa 2011** [46]	no	HD (*n* = 42) vs. OCD (*n* = 20) vs. HC (*n* = 36)	HD: 51.14 ± 8.33, OCD: 31.21 ± 11.80, HC: 47.00 ± 12.29	HD: F = 31, M = 11, OCD: F = 8, M = 12, HC: F = 29, M = 7	N/A	HD: 23; OCD: 22	CGI, ADIS-IV, HRS-I, HRSD-17, SIGH-D, SI-R, OCI-R, FIS	IGT, The dissonance reduction task	No significant differences between OCD and HC.	N/A
**Zhang et al., 2015** [6]	yes	umOCD (*n* = 57) vs. mOCD (*n* = 77) vs. rOCD (*n* = 48) vs. HC (*n* = 115).	umOCD: 28.07 ± 7.73, mOCD: 27.92 ± 7.07, rOCD: 28.50 ± 7.61, HC: 27.32 ± 7.81	OCD: F = 87, M = 95 [um = OCD: F = 30, M = 27, mOCD: F = 42, M = 35, rOCD: F = 23, M = 25], HC: F = 60, M = 55.	umOCD: 75.95 ± 45.69 months, mOCD: 65.83 ± 46.35 months, rOCD: 63.60 ± 36.73 months	overall, 125	SCID for DSM-IV criteria, HARS-14, HDRS-17, Y-BOCS	DST, TMT, WCST, IGT, GDT.	umOCD, mOCD and rOCD < HC at netscore and at the last three blocks.	DM deficits under ambiguity persisted regardless the medication status and symptom remittance.
**Zhang et al., 2015** [19]	no	OCD (*n* = 55) vs. UFDR (*n* = 55), HC (*n* = 55)	OCD: 26.51 ± 7.84, UFDR: 28.42 ± 7.37, HC: 27.85 ± 7.32	OCD: F = 33, M = 22, UFDR: F = 29, M = 26, HC: F = 31, M = 24	4.33 ± 3.66	all patients: medication-naïve	Clinical interview with the patient at minimum 1 relative (DSM-IV-TR criteria), Y-BOCS, HDRS-17, HARS	SCWT, DST, TMT, WCST, IGT, GDT, ToL.	OCD and UFDRs < HC	N/A
**Zhang et al., 2017** [41]	no	preop-OCD (*n* = 51) vs. postST-OCD (*n* = 24) vs. postLT-OCD (*n* = 32) vs. HC (*n* = 31)	preop-OCD: 30.71 ± 7.62, postST: 29.29 ± 5.72, postLT: 33.41 ± 7.86, HC: 37.77 ± 10.83	preop-OCD: F = 15, M = 36, postST-OCD: F = 8, M = 16, postLT: F = 12, M = 20, HC: F = 14, M = 17	preop-OCD: 10.66 ± 7.12; postST: 8.92 ± 3.55; postLT: 10.50 ± 4.85	N/A	MINI for DSM-IV-TR, Y-BOCS, HDRS-17, HARS	IGT	preopOCD < HC postLT-OCD = HC	N/A

Abbreviations: AD = Alcohol Dependence; ADHD = Attention Deficit Hyperactivity Disorder; ADIS-IV = Anxiety Disorders Interview Schedule for DSM-IV; AFLT = Associative feedback learning task; AN = anorexia nervosa; AnxD = Anxiety disorders; AUDIT = the Alcohol Use Disorders Identification Test; AVLT = Auditory Verbal Learning Test; BABS = the Brown Assessment of Beliefs Scale; BAI = Beck Anxiety Inventory; BD = bipolar disorders; BDI = Beck Depression Inventory; BDI-II = Beck Depression Inventory-II; BGT = The Bender Gestalt Test; BIS-11 = the Barratt Impulsiveness Scale-11; BN = bulimia nervosa; BVMT-R = the Brief Visual Memory Test-Revised; CBT = Cognitive-Behavioural Theraphy; CTT = Colour Trails Test; CFT = Complex Figure Test; CGI = the Clinical Global Impression; COWA = Controlled Oral Word Association; CY-BOCS = Children Yale-Brown Obsessive-Compulsive Scale; DM = Decision Making; DST = Digit Span test; ED = Eating Disorder; EDE-Q = Eating Disorders Examination Questionnaire; FIS = the Frost Indecisiveness Scale; FMPS = Frost Multidimensional Perfectionism Scale; fMRI = functional magnetic resonance imaging; FTND = Fagerstrom Test for Nicotine Dependence; GDT = Game of Dice Task; HARS = Hamilton Anxiety Rating Scale; HC = Healthy Controls; HD = Hoarding Disorder/Compulsive Hoarding; HDRS = Hamilton Depression Rating Scale; HE-OCD = High Expressive OCD: S- and/or L-allele of the the serotonin transporter gene act in a nearly dominant way; HRS = Hoarding Rating Scale-Interview; ICD-10 = International Statistical Classification of Diseases and Related Health Problems, 10th version; IGT = Iowa Gambling Task; LE-OCD = Low Expressive OCD: S- and/or L-allele of the the serotonin transporter gene did not act in a nearly dominant way; LPS-4 = the subtest 4 of the Leistungspruf-system; MADRS = the Montgomery–Åsberg depression rating scale; MINI = Mini-International Neuropsychiatric Inventory; MINI-DIS = Mini-International Neuropsychiatric Interview-Diagnostic Interview Schedule; mOCD = medicated OCD; mWCST = the modified Wisconsin Card Sorting Test; NART = National Adult Reading Test; NIMH-OCS = NIMH-Global Obsessive Compulsive Scale; nt-OCD = non-traumatic OCD; OAT = Object Alternation Test; OBQ-87 = the Obsessive Beliefs Questionnaire-87; OCD = Obsessive-Compulsive Disorder; OCI-R = The Obsessive-Compulsive Inventory-Revised; PD = Panic Disorder; PG-YBOC = Y-BOCS adapted for pathological gambling; PI = the Padua Inventory; PG = Pathological Gambling ; postLT-OCD = post-operative long term OCD patients (average 3 years after surgery); postST-OCD = post-operative short term (average 4 months after surgery); preop OCD = pre-operative OCD; prt-OCD = Pre-traumatic OCD (PTSD emerged before OCD); pst-OCD = Post-Traumatic OCD; PSWQ = The Penn State Worry Questionnaire; RCFT = Rey’s Complex Figure Test RCFT; rOCD = patients remitted from with OCD; RSPM = Raven Standard Progressive Matrices; SI-R = the Savings Inventory-Revised; SCID = the Structured Clinical Interview for DSM Disorders; SCR = skin conductance response; SCWT = Stroop ColorWord Test; SDQ = Strengths and Difficulties Questionnaire; SHAPS = Snaith-Hamilton Pleasure Scale; SKZ = Schizophrenia; SI-R = The 23-item Saving Inventory-Revised; SIGH-D = the Structured Interview Guide for the HRSD; SOGS = the South Oaks Gambling Screen; SRLT = the simple reversal learning task; SRTT = Serial Reaction Time Task; SSRTT = Stop Signal Reaction Time Task; STAI = State Trait Anxiety Inventory; TAU = Treatment as usual; TIB = Test di Intelligenza Breve (Italian version of the National Adult Reading Test); TMT = the Trail Making Test (A and B); ToH = Tower of Hanoy; ToL = Tower of London; UD = unipolar disorder; UFDR = unaffected first-degree relative; umOCD = unmedicated OCD; WAIS III = the Wechsler Adult Intelligence Scale; WASI = the Wechsler Abbreviated Scale for Intelligence; WASI—R = WASI revised; WCST = Wisconsin Card Sorting Test; WISC-III the Wechsler Intelligence Scale for Children-Third Edition; WMS III = Wechsler Memory Scale III; WMS-R LM = the Wechsler Memory Scale Logical Memory; Y-BOCS = Yale-Brown Obsessive-Compulsive Scale; Y-BOCS-SC = the Y-BOCS symptom checklist.

## Data Availability

Data will be shared upon request from any qualified investigator.

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
