# Peer review of "Obsessive-Compulsive Disorder and Decision Making under Ambiguity: A Systematic Review with Meta-Analysis"

_brainsci, 2021, doi:10.3390/brainsci11020143_

Round 1
Reviewer 1 Report
This very interesting meta-analysis study showed the importance of sub-endophenotypes like poor decision making in clinical applications in individuals with OCD.
Authors have included 16 studies with medicated and non medicated OCD patients performed gambling behavior task Iowa Gambling Task (IGT).They did find weaker decision making at completion of IGT in both medicated and unmedicated OCD patients. In conclusion, authors suggest decision making impairment might represent a potential endophenotype lying between the clinical manifestation of OCD and its neurobiological aetiology.
Overall, abstract, introduction, methods and results are well written and justify the context of the study and report results in organized way.
As earlier studies suggested involvement of several distinctive excecutive processes in performing the IGT task (A. Bechara, A. R. Damasio, H. Damasio, & S. W. Anderson, 1994) and effect of poor response inhibition on performance of IGT (Noël et al., 2007) in individuals with alcoholism. Therefore, I would like to suggest addition of prepotent response inhibition discrepancies in unmedicated OCD individuals below 18 years of age as an alternative mechanism behind the poor decision making in discussion section. Mancini et al. (2018) and Giovanni et al., (2020) reported the loss of proactive and reactive inhibition in children's with OCD symptoms in comparison of pure inhibitory TIC disorder. This suggest that there might be an impaired prepotent strategie making is involved in poor decision making of OCD patients.
Author Response
Response: we discusses the suggested topic as follows: “Finally, inhibitory control was shown to be impaired in unmedicated adolescent patients with OCD, and this impairment was correlated with the severity of their symptoms [60]. Inhibitory control is an executive function which allows adapting motor behavior according to the context in which the subject is embedded; in particular, reactive inhibition refers to the ability of the person to react to a stop signal, and it is usually evaluated through a Stop Signal Reaction Time Task (SSRTT); proactive inhibition consist in the ability of patients to shape their response strategies in anticipation of known task demands [60, 61]. Mancini and colleagues [60] found that the more severe were OCD symptoms, the more impaired were both proactive and reactive inhibition in in a group of adolescents with OCD (but not in an age-matched group of patient with Tourette syndrome). Decision making, and thus the IGT, strongly affected by one’s ability of response inhibition [63]. In our systematic review, only one paper implemented both the IGT and the SSRTT in a sample of adult with OCD and HC, and found that OCD performed worse than HC at the IGT but not at the SSRTT [30]; future studies investigating proactive inhibition in adults with OCD might help clarify whether this phenomenon occurs only in children and adolescent with OCD or could be generalized to the adult age.”. Relevant references were added.

Reviewer 2 Report
While I somewhat appreciate the rationale for the current review, in my opinion, this manuscript lacks a rigorous systematic review approach. There are many guides to systematic review and meta-analysis writing online that could help you.
Specific comments:
- "OCD and Decision Daking" - I believe you meant "Decision Making".
- "chronic doubting" - this is a rather awkward word choice/phrase.
- As per PRISMA guidelines, please specify in the methods section if the review protocol was prospectively registered. Indicate if a review protocol exists, if and where it can be accessed (e.g., Web address).
- When PubMed is used for the search, MESH terms are always recommended to be included. Please also provide the full electronic search strategy used to identify studies, including all search terms and limits for at least one database.
- It is unclear how many investigators were involved in the study selection and abstraction process and how were disputes regarding the inclusion/exclusion of studies resolved? The methods used were not adequately described; exactly who did what to identify, review, assess and resolve disagreements in the identified manuscripts. More details are required.
- How was the diagnosis of OCD made in the various studies reviewed? This should be specified in the data table.
- There was no risk of bias assessment; quality appraisal is central to any systematic review. To enhance the reproducibility and comparability of this review to future reviews of a similar topic (possibly an update of this review) I recommend including a risk of bias assessment using ROBINS-I, since it is the newest and most robust method of assessing risk of bias in systematic reviews/meta-analyses.
- Clinicians have long considered doubt to be a fundamental characteristic of OCD. However, the clinical relevance of doubt in OCD have not been addressed. In some studies, doubt was found to be strongly associated with global impairment of function and poor response to cognitive behavioral treatment and pharmacotherapy. This should be the area of focus, discussion and research.
Author Response
Reviewer 2
While I somewhat appreciate the rationale for the current review, in my opinion, this manuscript lacks a rigorous systematic review approach. There are many guides to systematic review and meta-analysis writing online that could help you.
Specific comments:
- "OCD and Decision Daking" - I believe you meant "Decision Making".
Response: we thank the reviewer and we corrected that typo.
2. "chronic doubting" - this is a rather awkward word choice/phrase.
Response: we rephrased it as “pathological doubt” in the abstract and in the main text.
3. As per PRISMA guidelines, please specify in the methods section if the review protocol was prospectively registered. Indicate if a review protocol exists, if and where it can be accessed (e.g., Web address).
Response: the study was not previously registered, which is why we did not provide an ID number or a web address.
4. When PubMed is used for the search, MESH terms are always recommended to be included. Please also provide the full electronic search strategy used to identify studies, including all search terms and limits for at least one database.
Response: strings inputted on PubMed for the bibliographic search, with relevant results emerged at the time of the search, are shown in Supplementary Material
5. It is unclear how many investigators were involved in the study selection and abstraction process and how were disputes regarding the inclusion/exclusion of studies resolved? The methods used were not adequately described; exactly who did what to identify, review, assess and resolve disagreements in the identified manuscripts. More details are required.
Response: We enriched the Method section with the requested information, as follows: “In September 2020, we conducted a systematic literature search in PubMed (http://www.ncbi.nlm.nih.gov) and EMBASE (https://www.elsevier.com/solutions/embase-biomedical-research) using the keywords “OCD” or “Obsessive-Compulsive Disorder” in combination with the keywords “Iowa Gambling Task” or “Gambling Task”. Strings inputted on PubMed, with respective results, are shown in Table 1. Additionally, the bibliographies of relevant articles were scanned for further suitable literature. Two authors (VN and ADA) screened first all abstracts, and then full-text articles independently. Disagreement was resolved by discussion between the two independent authors; if no agreement was reached, a third independent party (LR and BN) was involved as an arbiter”.
Moreover, we added as Supplementary Material the PRISMA Checklist for systematic review and meta-analysis.
6. How was the diagnosis of OCD made in the various studies reviewed? This should be specified in the data table.
Response: two column in Table 2 were added, with information about (i): diagnosis and psychiatric assessment, (ii) neuropsychological assessment.
7. There was no risk of bias assessment; quality appraisal is central to any systematic review. To enhance the reproducibility and comparability of this review to future reviews of a similar topic (possibly an update of this review) I recommend including a risk of bias assessment using ROBINS-I, since it is the newest and most robust method of assessing risk of bias in systematic reviews/meta-analyses.
Response: we thank the reviewer for his comment and suggestions. To the best of our knowledge, quality appraisal and risk of assessment bias are indicated and required when evaluating interventional studies e.g. RCT (see https://training.cochrane.org/handbook/current/chapter-25).
Specifically, when considering the ROBINS tool, most of the items would be “Not Applicable” in our study, making the results not valid and the process not useful. We are unaware of any applicable tools to our type of systematic review, but should the Reviewer recommend any appropriate ones, we would be fond to use it to make our SR more robust.
8. Clinicians have long considered doubt to be a fundamental characteristic of OCD. However, the clinical relevance of doubt in OCD have not been addressed. In some studies, doubt was found to be strongly associated with global impairment of function and poor response to cognitive behavioral treatment and pharmacotherapy. This should be the area of focus, discussion and research.
Response: We thank the reviewer for raising this important point. We have carefully evaluated this matter and improved our results and discussion accordingly as follow:
We have added in the Results:
“1.6 Pathological doubt
Only one study [27] directly evaluated the presence of a correlation between pathological doubt, as measured by the subscale “doubts about actions” of the Frost Multidimensional Perfectionism Scale, and the IGT score. They found that, only in OCD patients, IGT performance on the last two blocks was positively associated with the degree of doubts about actions.”
We have added in the Discussion:
“In fact, pathological doubt (i.e. a lack of certitude or confidence in one’s memory, attention, intuition, and perceptions, such that it is difficult to trust one’s internal experiences [48, 49]) is central in the clinical presentation of OCD. It underpins several symptoms, from “checking behaviors” (insufficient conviction about the completion of a task) to contamination concerns (insufficient conviction regarding the safety of a contacted object) [48], and this pervasive lack of certainty leads OCD patients towards a significant impairment in their daily life functioning. In addition to clinical observation, several experimental studies have investigated the chronic doubting and indecision characteristic of OCD [49; for a discussion see 48], which have been ultimately attributed to decision-making impairments [38, 50].”
Relevant references were added.
Finally, we have had our manuscript revised for English grammar and punctuation by a native English speaker.

Reviewer 3 Report
This is a generally very well written article. The topic is important and timely.
Data are accumulated rapidly and there is a need for careful and systematic evaluation of the collected data.
Addressing heterogeneity in OCD is a critical issue. I would be interested at more detailed analysis of the role of other than hoarding symptom dimensions, age of onset and gender.
Overall, I’d like to complement the authors on carefully done work.
Author Response
Reviewer 3
This is a generally very well written article. The topic is important and timely.
Data are accumulated rapidly and there is a need for careful and systematic evaluation of the collected data.
Addressing heterogeneity in OCD is a critical issue. I would be interested at more detailed analysis of the role of other than hoarding symptom dimensions, age of onset and gender.
Overall, I’d like to complement the authors on carefully done work.
Response: we did not add gender and age of onset in our meta-analysis because most of the studies did not provide specific IGT data (means, standard deviation and numerosity) for male and female separately. We added this as a limitation of our study: “First, several studies included in the systematic review could not be included in our meta-analysis, as they did not report the necessary statistical data; for the same reason, we could not run a meta-analysis evaluating a possible effect of gender and age of on-set, since most of the study did not investigate gender and age of onset effect and thus did not report the relevant statistical data”.
We added a paragraph (3.1.5 Gender) in the Results section of the Systematic review, as follows: “3.1.5 Gender: Most of the studies which investigated gender effect found neither a significant association between the IGT net score and gender, nor a difference in IGT performance between male and female subjects [15, 18, 32, 35]. Only Lawrence and colleagues [44] found a significant difference at the IGT performance between male and female participants, with 69% of men and 34% of women guessing the correct (or partially correct) strategy, but they found neither a difference between HC and OCD patients, nor an interaction effect between gender and diagnosis. Finally, Martoni and colleagues [40] found that, within patients scoring higher at the “Forbidden Thoughts” factor of the Y-BOCS, female patients at had a lower probability to provide correct answers in the IGT test than male patients.”

Round 2
Reviewer 2 Report
Thank you for the revisions. Specific comments: 1. The organisation and presentation for Table 1 could be greatly improved. It is way too cluttered at the moment and the IGT findings should be rephrased in a more succinct manner rather than lifted verbatim. 2. What does the column "medications" refer to? Is it the duration or types of medications the patients were on? Please be explicit. 3. Publication bias was not assessed. 4. Authors did not discuss or perform any sensitivity analyses. 5. In terms of the choice of medication, was one superior to another, say antidepressant vs antipsychotic? Some comments may be helpful.Author Response
We would like to thank the reviewer for their positive feedback and for the substantial and helpful comments which we hope have helped us to improve the manuscript. For clarity we have chosen to address each comment in order one-by-one.
- The organisation and presentation for Table 1 could be greatly improved. It is way too cluttered at the moment and the IGT findings should be rephrased in a more succinct manner rather than lifted verbatim.
Response: Thank you for the suggestion, we shortened the column with IGT findings to improve the table reading.
- What does the column "medications" refer to? Is it the duration or types of medications the patients were on? Please be explicit.
Response: the column “medications” refers to the number of patients, in each study, who were receiving oral medications (of any kind) prescribed for treating OCD at the time of testing. We changed the heading of the column with “Number of patients receiving medications”.
- Publication bias was not assessed.
Response: We added this paragraph in the method section: “The problem of publication bias (i.e. the existence of unpublished studies with negative results) was estimated informally by inspecting the funnel plot of effect size against standard error for asymmetry. We first run the analyses including the entire set of studies and then subsequently re-run them without the potential outliers identified based on visual inspection of the funnel plot”. Funnel plot was added to our paper (Figure 3). Moreover, the main meta-analysis was re-run after potential outliers (Figure 4). The following paragraph was added in the Results section: “The funnel plot (Fig. 3) was fairly symmetrical suggesting a low risk of publication bias. However, 5 studies overall accounting for 22.7% of the total weight in the meta-analysis were identified as possible outliers (Fig. 2, Fig. 3). A sensitivity analysis was performed after removal of these 5 studies, which confirmed a significant difference in the IGT between OCD and HC (SMD: - 0.64 95% C.I. [-0.76; -0.52]; z = 10.54; p < 0.001) (Fig. 4). In this case, however, the heterogeneity between studies dropped significantly (I-2 = 2%) and was no longer significant (chi-2 = 10.18; p = 0.42)”. Figure 4 was added and the other figures were renamed accordingly.
- Authors did not discuss or perform any sensitivity analyses.
Response: our sub-analysis (age and medications) are sensitivity analyses taking into account biological factors, which might have an impact on the outcome. Therefore, we renamed them in the paper as sensitivity analysis and only considered as sub-analysis the one addressing IGT trends in the 5 blocks. We added this paragraph in the Method section: “To explore reasons of heterogeneity, sensitivity analyses were performed considering two main biological factors potentially able to influence the outcome, i.e., in medicated and unmedicated patients, as well as in adults and in children (respectively above and below 18 years of age).” Moreover, sensitivity analysis without potential outliers was added (see our response to point 3).
- In terms of the choice of medication, was one superior to another, say antidepressant vs antipsychotic? Some comments may be helpful.
Response: We thank the reviewer for the comment, we haven’t taken into account the role of the specific type of medication. We agree that this is an important point and we had highlighted it amongst the study limitation, as follows: “finally, amongst medicated patients, it was not possible to further divide the sample according to the type of medication taken due to the small sample sizes, which might have create another bias”.